# Activated Alpha-2 Macroglobulin Improves Insulin Response via LRP1 in Lipid-Loaded HL-1 Cardiomyocytes

**DOI:** 10.3390/ijms22136915

**Published:** 2021-06-28

**Authors:** Virginia Actis Dato, Gustavo Alberto Chiabrando

**Affiliations:** 1Centro de Investigaciones en Bioquímica Clínica e Inmunología (CIBICI), Consejo Nacional de Investigaciones Científicas y Técnicas (CONICET), Córdoba 5000, Argentina; actis.dato@unc.edu.ar; 2Departamento de Bioquímica Clínica, Facultad de Ciencias Químicas, Universidad Nacional de Córdoba, Córdoba 5000, Argentina

**Keywords:** heart, lipoprotein, metabolism, glucose, LRP1

## Abstract

Activated alpha-2 Macroglobulin (α_2_M*) is specifically recognized by the cluster I/II of LRP1 (Low-density lipoprotein Receptor-related Protein-1). LRP1 is a scaffold protein for insulin receptor involved in the insulin-induced glucose transporter type 4 (GLUT4) translocation to plasma membrane and glucose uptake in different types of cells. Moreover, the cluster II of LRP1 plays a critical role in the internalization of atherogenic lipoproteins, such as aggregated Low-density Lipoproteins (aggLDL), promoting intracellular cholesteryl ester (CE) accumulation mainly in arterial intima and myocardium. The aggLDL uptake by LRP1 impairs GLUT4 traffic and the insulin response in cardiomyocytes. However, the link between CE accumulation, insulin action, and cardiac dysfunction are largely unknown. Here, we found that α_2_M* increased GLUT4 expression on cell surface by Rab4, Rab8A, and Rab10-mediated recycling through PI_3_K/Akt and MAPK/ERK signaling activation. Moreover, α_2_M* enhanced the insulin response increasing insulin-induced glucose uptake rate in the myocardium under normal conditions. On the other hand, α_2_M* blocked the intracellular CE accumulation, improved the insulin response and reduced cardiac damage in HL-1 cardiomyocytes exposed to aggLDL. In conclusion, α_2_M* by its agonist action on LRP1, counteracts the deleterious effects of aggLDL in cardiomyocytes, which may have therapeutic implications in cardiovascular diseases associated with hypercholesterolemia.

## 1. Introduction

Cardiovascular disease is frequently associated with metabolic syndrome, type 2-diabetes mellitus (T2DM) and obesity, and impaired insulin response is one of the main features in these patients [1,2,3]. The heart requires a large amount of energy for its contractile function and it is able to use different substrates [4]. The glucose uptake by the myocardium represents approximately 30% of the energy source available for this tissue [5,6,7]. In this way, insulin resistance has a marked influence on cardiac metabolism because the contribution of substrates is altered and there are drawbacks in the metabolic adaptations in the myocardium when the action of insulin is affected [8,9].

The alpha-macroglobulin family represents a well-defined group of proteinase inhibitors expressed in several vertebrate species, including human and mouse [10,11]. In human, two forms are identified, α_2_-macroglobulin (α_2_M) and pregnancy zone protein (PZP), which are analogues to tetrameric α_2_M and single chain murinoglobulin (MUG) in mouse, respectively [10,12]. Human and mouse α_2_M are structured by four identical 180-kDa subunits and can inhibit a broad spectrum of proteinases (serine-, metal-, aspartic-, and cysteine-proteinases). After α_2_M/proteinase interaction, a peptide bond localized within the bait region of the inhibitor molecule is cleaved. This event leads to a conformational change in the α_2_M, together with the trapping of the proteinase, activation, and cleavage of internal ß-cysteinyl-γ-glutamyl thiol esters, and exposure of carboxy terminal receptor recognition domains important for the fast clearance by a specific interaction with the cluster I/II of the α-subunit of LRP1 (for Low-density Lipoprotein Receptor-related Protein-1) [13,14]. This α_2_M-proteinase complex, also termed activated α_2_M or α_2_M*, can also regulate several cellular events such as cell proliferation and migration, through the activation of intracellular signalling pathways mediated by LRP1 [15]. A cardiac isoform of α_2_M (c-α_2_M) has been characterized in human and rats [16,17]. This c-α_2_M induces hypertrophic cell growth in ventricular cardiomyocytes via ERK1/2 and PI_3_K/Akt and improves cardiac cell function through its interaction with LRP1 [16,18,19]. Cardiac and liver α_2_M in humans and rats share a high homology [16]. Although a high level of plasma α_2_M has also been found in patients with insulin resistance in a large cohort, its role in this process is not clearly established yet [19].

LRP1 is a member of the LDL receptor gene family and is expressed in different cell types includes cardiomyocytes [15,20]. The α_2_M*/LRP1 interaction leads to lysosomal degradation of α_2_M* with an endocytic recycling of LRP1 to the plasma membrane (PM) [21]. This recycling involves different types of traffic, through a mechanism dependent on GTPases, such as Rab4 (short loop), from compartments of endocytic recycling (CRE) via Rab11 (long loop) and under certain conditions LRP1 sorting to PM mainly from preformed intracellular vesicles dependent on GTPases involved in the exocytic pathway, including Rab8A and Rab10 [21,22]. It has been demonstrated that LRP1 plays anti-inflammatory functions by selective suppression of toll-like receptor-4 (TLR-4) activity [23,24]. Based on this property, a group of LRP1 ligands including α_2_M* have been defined as agonist ligands since their can potentiate the anti-inflammatory action of this receptor [24]. By contrast, other ligands have pro-inflammatory effects, which are termed antagonist ligands of LRP1, such as RAP and lactoferrin [24]. In this way, it has been proposed LRP1 agonist ligands to provide cardioprotection during ischemia–reperfusion after acute myocardial infarction [25,26]. Other recent work has developed antibodies against CR8/CR9 domain in the cluster II of LRP1, which are crucial for the interaction of this receptor with atherogenic lipoproteins, to prevent foam cell formation and atherosclerosis development [27,28].

It is known that LRP1 also regulates the intracellular trafficking of insulin-responsive vesicles containing GLUT4, the main insulin-sensitive glucose transporter [29,30,31]. Moreover, LRP1 binds to insulin receptor (IR) and regulates intracellular insulin signaling in cardiomyocytes, neurons, and hepatocytes [20,32,33]. The cluster II of LRP1 also plays a critical role in the internalization of atherogenic lipoproteins, such as aggregated low-density lipoproteins (aggLDL), promoting intracellular cholesteryl ester (CE) accumulation mainly in arterial intima and myocardium [20,34]. In a previous work we have shown that LRP1-mediated uptake of aggLDL increased intracellular CE accumulation and impaired the insulin response in HL-1 cardiomyocytes by decreased insulin-induced intracellular signaling, glucose transporter type 4 (GLUT4) translocation to PM and glucose uptake [20], suggesting that aggLDL is an antagonist ligand of LRP1 in the regulation of insulin response.

Considering the evidence referenced above, we hypothesize that α_2_M* can potentiate the insulin response through its agonist action on LRP1, and also prevents the deleterious effect of aggLDL in HL-1 cardiomyocytes. Thus, in the present work we evaluate whether α_2_M* via LRP1 improves the insulin-induced response in CE-loaded HL-1 cardiomyocytes.

## 2. Results

### 2.1. α2M* Increases GLUT4 Expression on PM through PI_3_K/Akt and MAPK/ERK Activation Pathways

It was identified a cardiac isoform of α_2_M (c-α_2_M) in humans and rats, which resulted to be identical to liver α_2_M [16]. Thus, we evaluate whether cardiomyocytes may express the murine c-α_2_M in cells cultured under control conditions and in the presence of aggLDL. Previously we found that aggLDL produced a marked CE accumulation in HL-1 cardiomyocytes, which impair the insulin-induced response in these cells [20]. Through reduced SDS-PAGE and subsequent Western blot using a specific antibody against tetrameric α_2_M, we detected a 180 kDa subunit in cell lysates of HL-1 cardiomyocytes cultured in control conditions, indicative of c-α_2_M (Figure 1a,b). In the same figures it is shown that after 24 h of aggLDL treatment a significant increase of murine c-α_2_M was found. Thus, these data indicate that HL-1 cardiomyocytes expressed c-α_2_M and increased its expression under CE accumulation conditions by aggLDL.

In previous works we showed that α_2_M* via LRP1 activates PI_3_K/Akt or MAPK/ERK or both intracellular pathways depending on the cell types [21,35,36]. Herein, we evaluate the α_2_M* intracellular signaling activation in HL-1 cardiomyocytes by Western blot assays. Figure 1c–f shows that α_2_M* induced a significant phosphorylation of Akt (p-Akt) and ERK (p-ERK) at 5 min but not at 15 min of stimulus, indicating that α_2_M* promotes a fast and short activation of both intracellular pathways in this cell type.

In a previous work, we found that α_2_M* promotes LRP1 traffic to PM in the Müller glial-derived cell line, MIO-M1 cells [21]. Moreover, in this cell type we demonstrated that insulin increases the LRP1 expression on cell surface by a regulated exocytic pathway, involving the sorting of GLUT4-storage vesicles (GSVs) to the PM [37]. In myocardium, the insulin-induced intracellular signaling activation leads to GLUT4 translocation to the PM [20]. Considering these data, we analyze the α_2_M* effects on the GLUT4 traffic to the cell surface in HL-1 cardiomyocytes by biotin-labeling protein assays. Figure 2a,b show increased levels of GLUT4 on cell surface by α_2_M* at 5 min of stimulation with respect to the control condition. By contrast, α_2_M* stimulus for 15 min did not show effect of GLUT4 expression on PM compared to control. This evidence was corroborated by confocal microscopy in cells treated with α_2_M* for 5 min and then incubated with anti-GLUT4 in non-permeabilizing conditions (Figure 2c). Similar results were found for LRP1 and sortilin, other constitutive proteins of GSVs [29,37] (Figure 2d,e). Next, we evaluate whether GLUT4 traffic to the PM was dependent on PI_3_K/Akt and MAPK/ERK intracellular signaling activation by α_2_M*. For this, we performed a cell surface protein detection assay in which HL-1 cardiomyocytes were pre-incubated 30 min with wortmannin (PI_3_K inhibitor) or PD98059 (MAPK inhibitor) and then treated with α_2_M* for 5 min. Figure 2f,g shows that α_2_M*-induced GLUT4 increase on PM was significantly reduced through the blocking PI_3_K/Akt and MAPK/ERK intracellular signaling activation. Thus, our data shows that α_2_M* can activate PI_3_K/Akt and MAPK/ERK pathways and increase GLUT4 expression at the cell surface of HL-1 cardiomyocytes.

### 2.2. α2M* Promotes GLUT4 Endocytic Recycling through Rab4, Rab8A, and Rab10 GTPases

In previous work we have shown that α_2_M* promotes the exocytic route of LRP1 to PM by Rab10 GTPase activation [21]. Moreover, several reports found that GLUT4 traffic to PM induced by insulin is also dependent of Rab10 activation together with Rab8A [37,38,39,40]. Thus, our interest was to evaluate whether α_2_M* can activates the exocytic routes of GLUT4 to PM. For this, HL-1 cardiomyocytes were treated with α_2_M* for 5 min and then we analyzed the colocalization between GLUT4 and different markers of subcellular compartments, such as EEA1 for early endosomes, Rab4 for short loop endocytic recycling compartments, Rab11 for long loop endocytic recycling compartments, and Rab8A and Rab10 mainly associated with exocytic compartments. Figure 3 shows that α_2_M* increased the colocalization of GLUT4 in Rab4^+^, Rab8A^+^, and Rab10^+^ subcellular compartments in comparison with control conditions. By contrast, α_2_M* did not produce significant change in GLUT4 localization in early endocytic endosome (EEA1^+^) nor long loop endocytic recycling compartments (Rab11^+^) (Appendix A). Thus, these data suggest that α_2_M* promotes the GLUT4 traffic to PM by Rab4- dependent endocytic recycling and Rab8A and Rab10-exocytic route.

### 2.3. α2M* Enhances Insulin-Induced 2-NBDG Uptake

Considering the α_2_M* effect on the GLUT4 traffic to the PM, we evaluated the glucose uptake in HL-1 cardiomyocytes. Thus, cells were treated with α_2_M* for 2 to 30 min together with 2-NBDG, a glucose fluorescent analogue. Figure 4a,b shows that α_2_M* did not produce significant changes in 2-NBDG uptake with respect to the non-stimulated control. Thus, although α_2_M* was able to induce the GLUT4 traffic to PM, it had no effect on the glucose control. Considering that insulin induces GLUT4 traffic and the glucose uptake into different cell types, our interest was to evaluate if α_2_M* may have effect on the insulin response in HL-1 cardiomyocytes. For this, cells were treated with α_2_M* and insulin for 2 to 30 min together with 2-NBDG. Figure 4a shows that the combined treatment of α_2_M* and insulin at the first 5 min promptly enhanced the GLUT4 expression on the PM compared with the individual effect of α_2_M* or insulin. After this time, the combined treatment of α_2_M* and insulin produced a similar level of GLUT4 expression on the cell surface than insulin and major than α_2_M* alone. In the same way, α_2_M* + insulin induced a fast 2-NBDG uptake at the first 5 min of stimulus, which was similar with insulin alone after this time (Figure 4b). The area under the curve analysis shows that α_2_M* + insulin significantly increased the 2-NBDG uptake both at the first 5 min as well as at 30 min of stimulus compared to insulin alone (Figure 4c,d). Using simple linear regression analysis, we found that α_2_M* + insulin increased the uptake rate respect to insulin stimulus (Figure 4e,f). These results suggest that α_2_M* enhances the insulin response, increasing GLUT4 traffic to PM and insulin-induced glucose uptake in HL-1 cardiomyocytes.

### 2.4. α_2_M* Blocks aggLDL Intracellular Accumulation by LRP1

In a previous work we found that LRP1 is the main receptor responsible for the binding and endocytosis of aggLDL in HL-1 cardiomyocytes [20]. These lipoproteins bind to cluster II of α-subunit of LRP1, and this domain is also involved in the interaction with α_2_M* [13,14,20,34]. Other studies showed that anti-P3 antibodies reduced foam cell formation through the blockade of LRP1 interaction with atherogenic lipoproteins in cluster II [27,28]. Here, we evaluated whether α_2_M* blocks aggLDL accumulation. HL-1 cardiomyocytes were incubated with α_2_M* together with DiI-labeled aggLDL (aggLDL-DiI) for 8 h at 37 °C. Figure 5a,b shows that aggLDL-DiI was accumulated in HL-1 cardiomyocytes, but this accumulation was inhibited by α_2_M*. In normal conditions, CE is stored at a low proportion in lipid droplets [41]. However, excessive aggLDL uptake leads to CE accumulation in lysosomes [42]. In previous studies, we showed that LRP1 did not localize in lysosomes in different cell types [21,37,43]. Lysosomal degradation of α_2_M* leads to an endocytic recycling of LRP1 to the cell surface [21]. Here, we analyzed the intracellular localization of LRP1 after aggLDL stimulation by confocal microscopy. Figure 5c shows that aggLDL generated a localization of LRP1 in the degradation compartments, similar to late endosomes/lysosomes [LAMP^+^], with respect to α_2_M* stimulus and control condition. This effect of aggLDL on the distribution of LRP1 in degradation compartments was fully blocked by α_2_M*. These data indicate that α_2_M* inhibits the internalization of the aggLDL by LRP1 and the receptor transport into degradation compartments.

### 2.5. α2M* Counteracts the Impairment of GLUT4 Traffic to PM and Glucose Uptake Induced by aggLDL

Previously, we found that aggLDL/LRP1 interaction leads to intracellular CE accumulation and impaired insulin response in HL-1 cardiomyocytes [20]. Here, we evaluated whether α_2_M* restores GLUT4 traffic to PM and glucose uptake impaired by aggLDL in these cells. For this, cells were treated with aggLDL, α_2_M* or both combined ligands for 8 h, then stimulated with insulin for 30 min, and the expression levels of GLUT4 on the cell surface were analyzed by biotin-labeling cell surface protein assays. As we previously found, in Figure 6a,b it is shown that aggLDL affected insulin-induced GLUT4 expression on the PM (Line 4), whereas α_2_M* did not modify the insulin effect at this time (Line 8). Moreover, α_2_M* alone induces the GLUT4 traffic to PM (Line 7) while aggLDL did not promote this effect (Line 3). Although the combination of α_2_M* and aggLDL did not produce significant changes in the GLUT4 levels in the cell surface (Line 5), α_2_M* counteracted the aggLDL deleterious effect, since it helped to promote GLUT4 translocation to cell surface by insulin (Line 6). The presence of α_2_M*, aggLDL, and both combined ligands induced the LRP1 expression on PM, which are in agreement with previous works [20,21,37]. Finally, we evaluated insulin-induced glucose uptake by HL-1 cardiomyocytes exposed to aggLDL and α_2_M*. Figure 6c,d shows that neither α_2_M*, aggLDL nor both combined ligands have any effects in 2-NBDG uptake by HL-1 cardiomyocytes. However, α_2_M* restored the insulin-induced 2-NBDG uptake abrogated by aggLDL with respect to the controls in these cells. These results indicate that α_2_M* improves the deleterious effects of aggLDL on the insulin-induced response in HL-1 cardiomyocytes.

### 2.6. α2M* Prevented aggLDL-Induced Cardiac Damage

Galectin-1 and 3 (Gal-1 and Gal-3) are the strongest predictors of cardiac damage and heart failure [44,45]. Our interest was to study whether aggLDL, α_2_M*, and both combined ligands may affect galectins expression in HL-1 cardiomyocytes. Figure 7a,b show that aggLDL increased the Gal-1 and Gal-3 mRNA expression in HL-1 cardiomyocytes, while this effect was counteracted by α_2_M*. These results suggest that α_2_M* also prevents aggLDL-induced cardiac damage associated with myocardial dysfunctions.

## 3. Discussion

It is well established that LRP1 acts as a scaffold protein for the insulin receptor (IR), which is considered critical for the intracellular insulin signaling activation in cardiomyocytes, neurons, and hepatocytes [20,32,33]. In the present study we showed that α_2_M*, mediated by its interaction with LRP1, enhanced the insulin response, increasing insulin-induced GLUT4 traffic to PM and the glucose uptake in HL-1 cardiomyocytes. By contrast, it has also been demonstrated that aggLDL, through its binding to LRP1, induces CE accumulation and impairs insulin-induced IR activation in this cell type [20]. Here, we found that α_2_M* counteracted the antagonist effect of aggLDL on LRP1, blocking the lipid loading induced by this modified lipoprotein and improved the insulin response in HL-1 cardiomyocytes. In addition, α_2_M* was able to prevent aggLDL-induced cardiac damage characterized by a decrease in Gal-1 and Gal-3 expression in HL-1 cardiomyocytes.

In previous works, an increased level of a cardiac form of α_2_M (c- α_2_M) in patients with insulin resistance was found [46,47], which was expressed in human and rat hearts [16,17]. Through its interaction with LRP1, c-α_2_M improves cardiac cell function and induces hypertrophic in ventricular cardiomyocytes via ERK1/2 and PI_3_K/Akt activation [16,18,19]. Here, we show that mouse c-α_2_M, the analogue form of human α_2_M, is also expressed in HL-1 cardiomyocytes, which is significantly increased in cells treated with aggLDL. This c-α_2_M could be secreted as a native form and activated by extracellular proteinases in cardiac tissue, which is then recognized and internalized by LRP1. However, it would also be interesting to know if c-α_2_M may be a product of extravasation from systemic α_2_M or only by local production at the cardiac level, and if its expression can be inducible by the aggLDL itself. In this way, α_2_M* could have an agonist effect on the LRP1 function counteracted the impairment in the insulin response induced by lipid accumulation in cardiomyocytes.

In previous studies, we found that α_2_M* activates PI_3_K/Akt or MAPK/ERK or both intracellular pathways depending on cell types [21,36]. It is known that these intracellular pathways mediate the LRP1 intracellular traffic to PM in MIO-M1 cells induced by α_2_M* or insulin [21,37]. In addition, PI_3_K/Akt and MAPK/ERK activation by insulin also leads to GLUT4 translocation to the PM [20,31]. Our present results showed that α_2_M* induced phosphorylation of Akt and ERK at short times of stimulus, suggesting that α_2_M* promotes a quick activation of both intracellular pathways in HL-1 cardiomyocytes. Then, we found that α_2_M* increased GLUT4 levels on the cell surface mediated by PI_3_K/Akt and MAPK/ERK intracellular signaling activation. Moreover, α_2_M* also increased sortilin levels at the PM, which is a constitutive protein of GSVs together with GLUT4 and LRP1 [37]. The Rab small GTPases play a critical role in the GLUT4 traffic to PM induced by insulin [31,37]. Here, we found that the α_2_M*-induced GLUT4 traffic involved the participation of Rab4, Rab8A and Rab10 GTPases from short loop endocytic route and exocytic pathway. Increased expression of GLUT4 on PM by α_2_M* significantly enhance insulin-induced glucose uptake, which was higher than insulin alone in the first 5 min of stimulation. However, α_2_M* alone did not produce significant glucose input respect to the non-stimulated control. This differential effect of α_2_M* with respect to insulin on the glucose uptake may be related to the capability of insulin to activate not only intracellular pathways such as PI_3_K/Akt and MAPK/ERK but also different proteins involved in GSVs retention inside the cell, tethering and fusion of GSVs to PM and activity of GLUT4, which α_2_M* would not activate [31,48]. In this way, it has been demonstrated that insulin stimulates endocytosis and exocytosis of GLUT4 through the participation of several molecular mediators such as ESYT1 and TC10 phosphorylated by Cdk5 [49,50]; MYO1C and MYO5 phosphorylated by CamKII [51,52]; and RIP140 and SEC5 phosphorylated by PKC [53,54]. Further studies are needed to explore if α_2_M* plays some function during the endocytosis and exocytosis of GLUT4. Nevertheless, the combined stimulation of α_2_M* and insulin showed a significant enhancement in the insulin response, characterized by an increased GLUT4 traffic to PM and insulin-induced 2-NBDG uptake in HL-1 cardiomyocytes. This improvement of the insulin response may be mediated through the enrichment of GLUT4 in the cell surface induced by α_2_M*, enabling a major number of glucose transporters available in the cells stimulated with insulin. Another possibility is that α_2_M* increases the endocytic recycling of LRP1, which could facilitate the molecular association between LRP1 and IR [20] and potentiate the IR activation induced by insulin. This evidence suggest that α_2_M* would have an important role as an insulin sensitizing agent in the heart, as proposed by other studies about patients with metabolic disturbances and heart failure [55].

In addition to α_2_M*, LRP1 can interact with different ligands with more or less affinity [15]. One of these ligands is aggLDL, which generates mainly CE uptake in cardiomyocytes, hypercholesterolemia being a promoting factor for cardiac dysfunction [20,56,57]. The aggLDL is recognized by a sequence located on the CR8/CR9 domain in the cluster II of the extracellular alpha chain of LRP1, and this interaction is mediated by heparin sulfate proteoglycans (HSPG) [15,27,28]. While α_2_M* interacts directly with LRP1 by clusters I/II with high affinity, which could represent a steric impediment to the binding of aggLDL [13,14,20,34]. In previous studies, we found that aggLDL promotes the molecular dissociation between LRP1 and IR with impairment of insulin-induced IR activation [20]. In this sense, our results showed that α_2_M* prevented aggLDL intracellular accumulation by LRP1 and the anomalous aggLDL-induced LRP1 localization in lysosomes. Moreover, α_2_M* counteracted the GLUT4 trafficking to PM and glucose uptake affected by aggLDL. In contrast to the agonist effect of α_2_M*/LRP1 on insulin response, aggLDL would be an antagonist ligand of LRP1 in this function in HL-1 cardiomyocytes. These results could underlie the link between intracellular CE accumulation in cardiomyocytes and cardiac insulin signaling abnormalities.

Finally, α_2_M* also prevented aggLDL-induced cardiac damage evidenced by a decreased level of Gal-1 and Gal-3, key mediators of cardiac lipotoxicity and heart failure [44,45]. Recent works have proposed LRP1 as an ideal target to prevent myocardial dysfunction [25,26,27,28]. In this way, it has been demonstrated that a short peptide, termed SP16, derived from serine-protease inhibitors (SERPINs), interacts with LRP1 and produces an agonist effect on the reduction of myocardial injury and preservation of cardiac systolic function in experimental acute myocardial infarction (AMI) [25,26]. On the other hand, by the use of antibodies against the CR8/CR9 domain in the cluster II of LRP1, foam cell formation and atherosclerosis development was prevented [27,28]. Further studies are needed to know if the use of α_2_M* may alter other processes in the heart such as inflammation or cardiac remodeling [25,26].

In conclusion, α_2_M* is an agonist ligand of LRP1 improving the insulin response in lipid-loaded HL-1 cardiomyocytes. Figure 8 is a schematic representation of α_2_M* counteracting the antagonist effect of aggLDL on LRP1, improving the insulin response characterized by PI_3_K/Akt and MAPK/ERK signaling activation as well as GLUT4 translocation to PM and glucose uptake in cardiomyocytes. All our findings may have important therapeutic implications in relation to the role of α_2_M* in the cardiac insulin response associated with hypercholesterolemia.

## 4. Materials and Methods

### 4.1. HL-1 Cardiomyocyte-Derived Cell Line, Cultures and Reagents

The murine HL-1 cardiomyocyte-derived cell line was generated by Dr. W.C. Claycomb (Louisiana State University Medical Centre, New Orleans, LA, USA). These cells showed cardiac characteristics similar of adult cardiomyocytes [58]. HL-1 cardiomyocytes were maintained in Claycomb Medium (Sigma-Aldrich, St. Louis, MO, USA) supplemented with 10% fetal bovine serum (FBS) (Invitrogen, Buenos Aires, Argentina), 100 M nor-epinephrine (Sigma-Aldrich, St. Louis, MO, USA), 100 units/mL penicillin, 100 g/mL streptomycin (Invitrogen), and L-glutamine 2 mM (GlutaMAX from Invitrogen, Buenos Aires, Argentina) in plastic dishes, coated with 12.5 g/mL fibronectin (Sigma-Aldrich) and 0.02% gelatin in a 5% CO_2_ atmosphere at 37 °C. The insulin solution from bovine pancreas (#I0516), Wortmannin and PD98059 were from Sigma-Aldrich. Rabbit anti-Akt (#9272), anti-ERK1/2 (#4695) and rabbit anti-pERK (#9101, Thr202/Tyr204) antibodies were from Cell Signaling Technology (Beverly, MA, USA). Rabbit anti-pAkt (Ser473, #07-789) antibody was from Merck (Darmstadt, Germany). Mouse anti-β-actin (#A2228) was from Sigma-Aldrich. Mouse anti-APT1A1 (#M7-PB-E9) was from ThermoFisher Scientific (Waltham, MA, USA). Rabbit anti-LRP1 (#ab92544), rabbit anti-GLUT4 (#ab654), mouse anti-GLUT4 (#ab 48547), rabbit anti-sortilin (#ab16640), rabbit anti-EEA1 (#ab2900), rabbit anti-Rab4 (#ab13252), mouse anti-Rab10 (#ab104859), rabbit anti-Rab11 (#ab65200), mouse anti-Rab8A (#ab128022), rabbit anti-mouse tetrameric α_2_M (#ab58703) and rabbit anti-LAMP1 (#ab208943) antibodies purchased from Abcam (Cambridge, MA, USA). Immunofluorescences were performed with secondary antibodies raised in goat IgG conjugated with Alexa Fluor 647, 594 or 488, (diluted 1/800) (Invitrogen, Buenos Aires, Argentina). The α_2_M was purified from human plasma following a procedure previously reported [59] and α_2_M* was generated by incubation with 200 mM methylamine–HCl for 6 h pH 8.2, as previously described [60].

### 4.2. LDL Isolation and Modification

LDL (d 1.019–d 1.063 g/mL) was isolated by ultracentrifugation using KBr gradients, in the density range 1.019–1.063 g/mL, from pools of plasma of normocholesterolemic volunteers. A Pierce kit (#23225, ThermoFisher Scientific (Rockford, IL, USA) was used for the protein quantification. Apolipoprotein B protein integrity was tested by SDS-PAGE in 10% acrylamide gels. Aggregated LDL was obtained by vortexing LDL in PBS 1X for 5 min at room temperature [20]. AggLDL was suspended in PBS 1X to a protein concentration of 100 µg/mL.

### 4.3. DiI-Staining of LDL

DiI (1,1-dioctadecyl-3,3,3,3-tetramethylindocarbocyanine, Invitrogen) is a lipophilic dialkylcarbocyanine that binds to lipoproteins and emits fluorescence at 565 nm. The LDL (100 µg/mL) was incubated with DiI in a proportion of 3 μL per 1 mg of lipoprotein in PBS 1X at 37 °C for 12 h and was then exhaustively dialyzed in PBS 1X for 24 h to eliminate the free DiI and filtered through a 0.22 μm filter [20]. Finally, DiI-LDL was aggregated mechanically by vortexing.

### 4.4. Western Blot Analysis

Cell protein extracts were prepared using RIPA lysis buffer (50 mM Tris–HCl pH 8.0, 150 mM NaCl, 1% Triton X-100, 0.5% sodium deoxycholate, 0.1% SDS, 1 mM PMSF, 10 mM sodium ortho-vanadate, and protease inhibitor cocktails (Sigma-Aldrich, St. Louis, MO, USA)). Forty micrograms of protein extracts were separated by electrophoresis on 10% SDS-polyacrylamide gels [61] and transferred to a nitrocellulose membrane [62] (GE Healthcare Life Science, Amsterdam, The Netherlands). Nonspecific binding was blocked with 5% non-fat dry milk in a Tris-HCl-0.01% Tween 20 (TBS-T) buffer for 60 min at room temperature. The nitrocellulose membranes were incubated overnight at 4 °C with primary antibodies, and secondary antibodies raised in goat anti-mouse IgG 680CW and goat anti-rabbit IgG 800CW (LI-COR Biosciences, Lincoln, NE, USA) diluted 1/10,000 for 1 h at room temperature. The specific bands were developed using Odyssey CLx near-infrared fluorescence imaging system (LI-COR) and were quantified by densitometric analysis using Image Studio Software (LI-COR).

### 4.5. Confocal Microscopy

The procedures were followed, as previously described [21,22]. Briefly, HL-1 cardiomyocytes were cultured on cover glass. After different stimulus the cells were washed with PBS 1X, fixed with 4% paraformaldehyde (PFA), quenched with 50 mM NH_4_Cl, permeabilized for 30 min with 0.5% (*v*/*v*) saponin and blocked with 2% bovine serum albumin (BSA) [20]. Primary antibodies (diluted from 1/100 to 1/250) were used for 1 h, and then secondary antibodies conjugated with Alexa Fluor (1/800) and Hoechst 33,258 colorant (1/2000) were used for 1 h. Finally, cells were mounted on glass slides with Mowiol 4–88 reagent from Calbiochem (Merck, Darmstadt, Germany). Fluorescent images were obtained with an Olympus FluoView FV1200 confocal microscope (Olympus, New York, NY, USA). Optical sections of the cells were obtained in 0.25-μm steps perpendicular to the *z*-axis. Images being processed using the FV10-ASW Viewer 3.1 (Olympus, New York, NY, USA) and quantified by ImageJ software. For microscope quantification of the colocalization level, a JACoP plug-in from ImageJ software (National Institutes of Health, New York, NY, USA) was used [63].

### 4.6. Biotin-Labeling Cell Surface Protein Assay

To determine the level of proteins at the cell surface we used a biotin-labeling protein assay (EZ-Link™ Sulfo-NHS-SS-Biotin [cat: 21331], Thermo Scientific, Rockford, IL, USA) as was previously described [20]. After stimulus, cells were incubated with a 0.12 mg/mL of biotin solution for 2 h at 4 °C, and then with 0.1 mM glycine solution for 30 min. The biotinylated proteins were pulled down using streptavidin-conjugated agarose beads (Pierce™ Streptavidin Agarose [cat: 20353], Thermo Scientific) for 2 h at room temperature. The biotinylated-plasma membrane proteins were eluted, then treated for Western blot and the nitrocellulose membranes were incubated with primary antibodies overnight at 4 °C and secondary antibodies were raised in goat anti-mouse IgG 680CW and goat anti-rabbit IgG 800CW (LI-COR) diluted 1/10,000 for 1 h at room temperature. The specific bands were developed using Odyssey CLx fluorescence imaging system (LI-COR) and were quantified by densitometric analysis using Image Studio Software (LI-COR). As the loading control of PM protein, total protein biotinylated-ATP1A1 and β-actin were used, respectively. Each biotinylated protein in the PM was related to biotinylated-ATP1A1 protein.

### 4.7. Cell Surface Protein Detection Assay

The procedure was followed, as previously described [37]. Briefly, after different stimulus cells were rinsed with cold PBS 1X, fixed with 4% (*v*/*v*) PFA, washed with 0.1 mM glycine, and blocked with 5% (*v*/*v*) horse serum for 30 min on ice. Cells were incubated with anti-GLUT4 (1/1000) and anti-ATP1A1 (1/2000) for 1 h on ice, followed by incubation with goat anti-rabbit IgG 800CW and goat anti-mouse IgG 680CW (LI-COR) secondary antibodies (1/10,000) for 1 h on ice. Fluorescence intensity was measured using the Odyssey CLx fluorescence imaging system (LI-COR) and quantified by densitometry using Image Studio Software (LI-COR). The content of GLUT4 in the PM was related to ATP1A1 protein. For other experiments, before stimulus, cells were pre-incubated for 30 min with 40 µM wortmannin or PD98059.

### 4.8. 2-NBDG Uptake Assay

Cells were incubated with 80 μΜ of 2-Deoxy-2-[(7-nitro-2,1,3-benzoxadiazol-4-yl) amino]-D-glucose (2-NBDG solution; Sigma-Aldrich) for 30 min together with different stimulus [20]. Then, cells were washed with PBS 1X, fixed with 4% PFA, quenched with 50 mM NH4Cl, permeabilized for 30 min with 0.5% (*v*/*v*) saponin, blocked with 2% BSA and incubated with Hoechst 33,258 colorant (1/2000) for 1 h. Finally, cells were mounted on glass slides with Mowiol 4–88 reagent. Fluorescent images were obtained with an Olympus FluoView FV1200 confocal microscope (Olympus, New York, NY, USA). Optical sections of cells were obtained in 0.25 μm steps perpendicular to the *z*-axis. Images were processed using the FV10-ASW Viewer 3.1 (Olympus, New York, NY, USA) and the total fluorescence in the whole cell area was quantified by ImageJ software.

### 4.9. Real Time-PCR

The cells were exposed to the different stimuli and treated with the TRIzol^®^ reagent (Invitrogen, Buenos Aires, Argentina) for total RNA extraction. One μg/20 μL of RNA was reverse transcribed using random hexaprimers and reverse transcriptase. The primers listed below were used to quantify the transcripts of Gal-1, Gal-3 and β-actin. The results were normalized to RT-PCR products of β-actin transcripts. For quantification we used real-time qRT-PCR (ABI 7500 Sequence Detection System, Applied Biosystems, Foster City,CA) and Sequence Detection software v1.4. Amplification conditions involved a warm start at 95 °C for 10 min, followed by 40 cycles at 95 °C for 15 s and 60 °C for 1 min. Relative gene expression was calculated by the 2-Ct method. Samples was analyzed in triplicate. No amplification was observed using water or RNA samples incubated without reverse transcriptase during cDNA synthesis.

Sequences of mouse primers:

Gal-1

forward: TCAGCCTGGTCAAAGGTGAT

reverse: TGAACCTGGGAAAAGACAGC

Gal-3

forward: CAGGAAAATGGCAGACAGCTT

reverse: CCCATGCACCCGGATATC

β-Actin

forward: AAATCTGGCACCACACCTTC

reverse: GGGGTGTTGAAGGTCTCAAA

### 4.10. Statistical Treatment of Data

For Western blot and cell-surface protein detection assay, the data was expressed as Mean ± SEM and comparisons between groups were analyzed by one-way ANOVA followed by Dunnett’s post-hoc analysis (GraphPad Prism 5.0, San Diego, CA, USA) or Student’s *t*-test. For confocal microscopy, at least 20 cells/condition were analyzed and data was expressed as Mean ± SD. The averages of the vesicle percentages containing colocalized proteins were calculated using the Manders’ coefficients. Comparisons between groups were analyzed by one-way ANOVA followed by Dunnett’s post-hoc analysis (GraphPad Prism 5.0) or Student’s *t*-test. Values of *p* < 0.05 were considered to be significant.

## Figures and Tables

**Figure 1 ijms-22-06915-f001:**
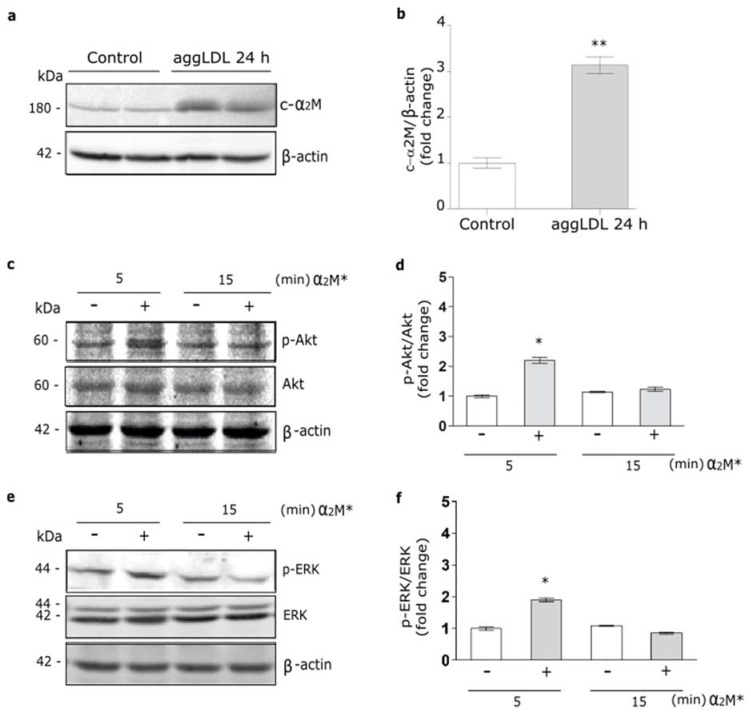
α_2_M* activates PI_3_K/Akt and MAPK/ERK pathways in HL-1 cardiomyocytes. (**a**) Western blot assay for the analysis of 180 kDa-subunit from mouse tetrameric α_2_M (c-α_2_M) expression in protein extract of HL-1 cardiomyocytes treated with aggLDL 100 µg/mL for 24 h. Bands are shown in duplicates. β-actin was used as loading control. The electrophoresis conditions are detailed in the Materials and Methods section. (**b**) Densitometric quantification of Western blot data for c-α_2_M/β-actin expressed as fold change respect to non-stimulated control (white bar). Values are expressed as mean ± SEM. ** *p* < 0.01 vs. non-stimulated control. (**c**,**e**) Western blot assay for the analysis of phosphorylated Akt (p-Akt) (**c**) or phosphorylated ERK (p-ERK) (**e**) in cells treated with α_2_M* 60 nM for 5 and 15 min. Total Akt (Akt), total ERK (ERK) and β-actin were used as loading control (**d**,**f**). Densitometric quantification of Western blot data for p-Akt/Akt (**d**) or p-ERK/ERK (**f**) expressed as fold change respect to non-stimulated control (white bar). Values are expressed as mean ± SEM. * *p* < 0.05 vs. non-stimulated control. Three experiments were performed (*n* = 3).

**Figure 2 ijms-22-06915-f002:**
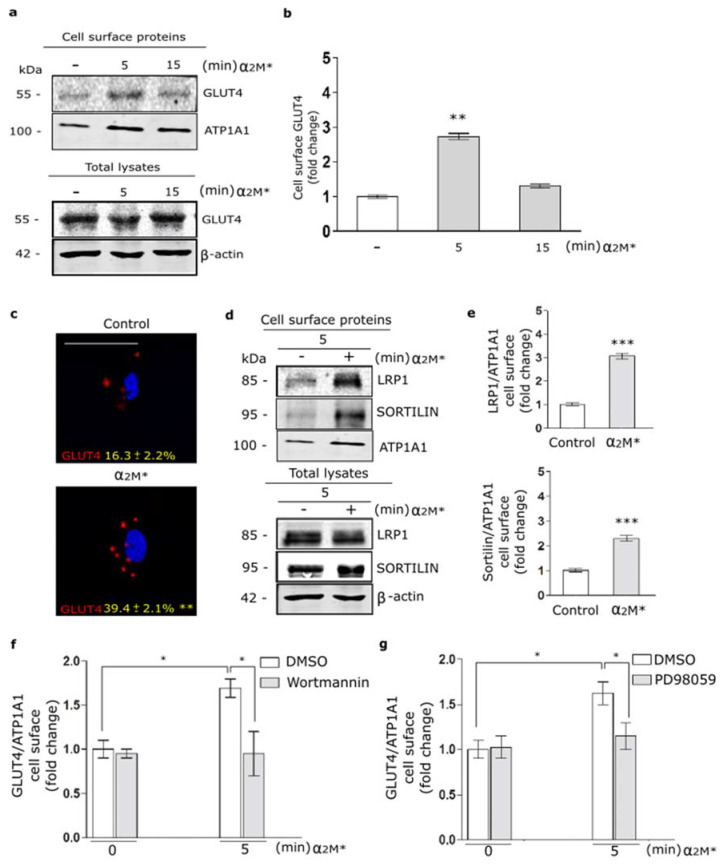
α_2_M* increases GLUT4 on cell surface by PI_3_K/Akt and MAPK/ERK signaling activation in HL-1 cardiomyocytes. (**a**) Biotin-labeling protein assay to measure GLUT4 level in the PM of cells stimulated with α_2_M* (60 nM) for 5 and 15 min at 37 °C. Biotin-labeled proteins were isolated with streptavidin-conjugated beads and then analyzed by Western blot for GLUT4 expression. ATP1A1 and β-actin were used as protein loading controls. (**b**) Densitometric quantification of Western blot data for surface biotin-GLUT4 related to biotin-ATP1A1 expressed as fold change with respect to the non-stimulated control (white bar). Values are expressed as mean ± SEM. ** *p* < 0.01 vs. non-stimulated control. Three experiments were performed (*n* = 3). (**c**) Confocal microscopy of HL-1 cardiomyocytes treated with α_2_M* (60 nM) for 5 min at 37 °C. Representative images of immunostained GLUT4 (red) in non-permeabilized cells (Surface). Quantitative analysis of fluorescence intensity per cell area are expressed as mean ± SD (%). ** *p* < 0.01 vs. non-stimulated control. Scale bar = 10 µm. At least 20 cells per condition were analyzed (*n* = 20). (**d**) Biotin-labeling protein assay was used to measure LRP1 and sortilin protein levels in the PM of cells stimulated with α_2_M* (60 nM) for 5 min at 37 °C. Biotin-labeled proteins were isolated with streptavidin-conjugated beads and then analyzed by Western blot for LRP1 and sortilin. ATP1A1 and β-actin were used as protein loading controls. (**e**) Densitometric quantification of Western blot data for surface LRP1 and sortilin related to ATP1A1 expressed as fold change with respect to the non-stimulated control (white bar). Values are expressed as mean ± SEM. *** *p* < 0.001 vs. non-stimulated control. Three experiments were performed (*n* = 3). (**f**,**g**) Cell surface protein detection assays to measure cell surface GLUT4 in cells pretreated with Wortmannin (40 µM) (**f**) or PD98059 (40 µM) (**g**) for 30 min and treated with α_2_M* (60 nM) for 5 min at 37 °C. The cell surface level of GLUT4 was analyzed in non-permeabilized cells using anti-GLUT4 antibody. Values are expressed as mean ± SEM. * *p* < 0.05 vs. indicated conditions. Three experiments were performed (*n* = 3).

**Figure 3 ijms-22-06915-f003:**
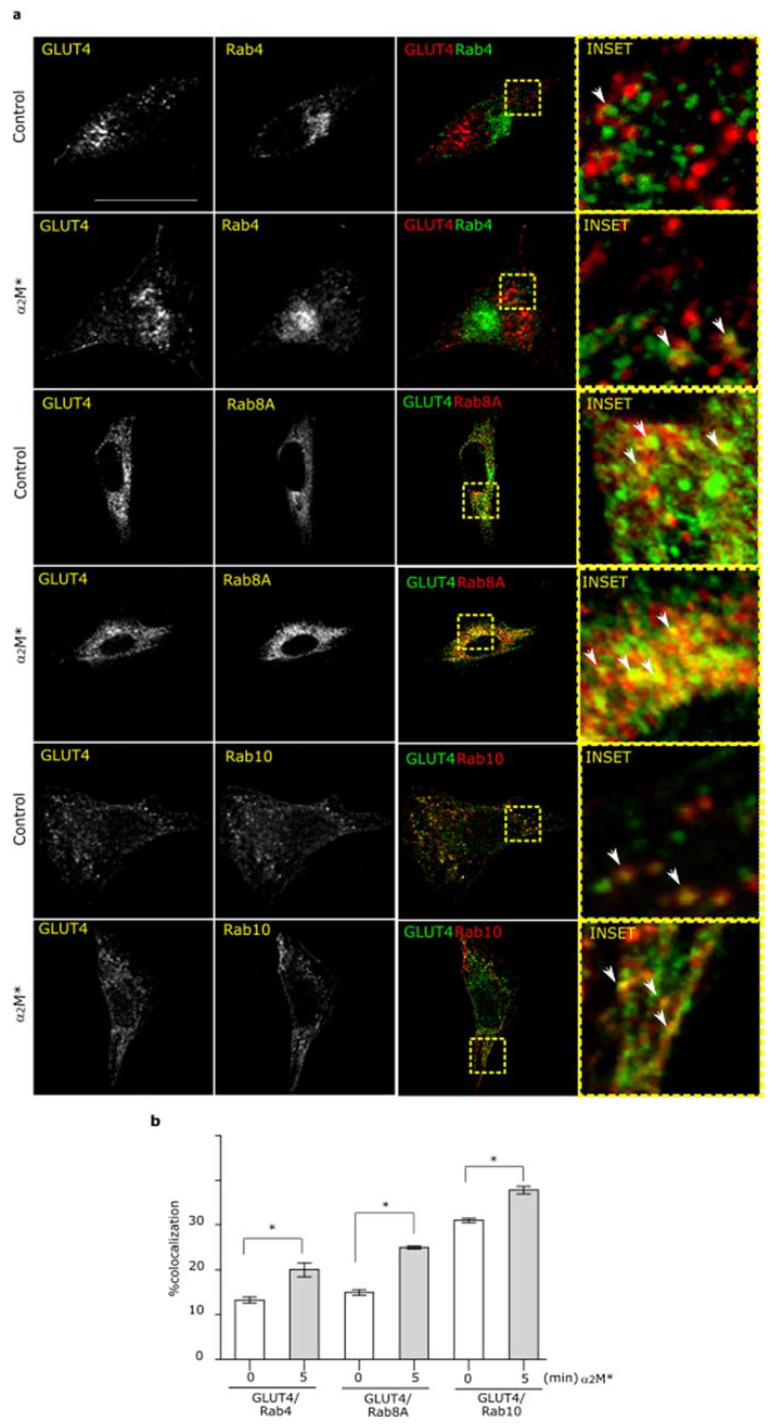
α_2_M* promotes endocytic and exocytic recycling of GLUT4 in HL-1 cardiomyocytes. (**a**) Confocal microscopy in cells treated with α_2_M* (60 nM) for 5 min at 37 °C. Images show colocalization between GLUT4 and Rab4 (green), Rab8A (red) or Rab10 (red). INSET represents magnification 4× of framed regions in dotted lines. White arrowheads indicate colocalization sectors. Scale bar = 10 µm. (**b**) Quantitative analysis of colocalization between GLUT4 and different markers by Manders’ coefficients expressed as mean ± SD (%). * *p* < 0.05 vs. indicated conditions. At least 20 cells per condition were analyzed (*n* = 20).

**Figure 4 ijms-22-06915-f004:**
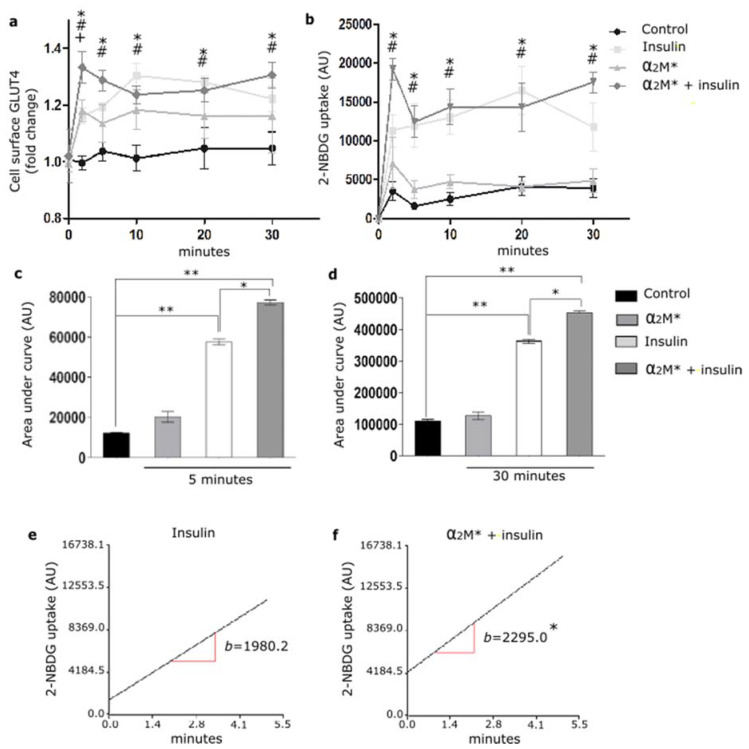
α_2_M* improves 2-NBDG uptake in HL-1 cardiomyocytes. (**a**) Cell surface protein detection assay to measure GLUT4 in cells treated with α_2_M* (60 nM), insulin (100 nM) or combined (α_2_M* + insulin) for different times at 37 °C. The cell surface level of GLUT4 was analyzed in non-permeabilized cells using anti-GLUT4 antibody. Values are expressed as mean ± SEM. + indicates significant differences between α_2_M* and control (*p* < 0.05). # indicates significant differences between insulin and control (*p* < 0.05). * indicates significant differences between α_2_M* + insulin and control (*p* < 0.05). Three experiments were performed (*n* = 3). (**b**) Uptake assay to measure 2-NBDG uptake by confocal microscopy in cells treated with α_2_M* (60 nM), insulin (100 nM) or combined (α_2_M* + insulin) for different times together with 2-NBDG 80 µM at 37 °C. Graph represents mean ± SEM of the fluorescence intensity of 2-NBDG per cell area. # indicates significant differences between insulin and control (*p* < 0.05). * indicates significant differences between α_2_M* + insulin and control (*p* < 0.05). Three experiment were performed (*n* = 3). (**c**,**d**) Area under curve quantified from (**b**) at 5 and 30 min, respectively. AU, arbitrary unit. Values are expressed as mean ± SEM. * *p* < 0.05 vs. indicated conditions. ** *p* < 0.01 vs. indicated conditions. (**e**,**f**) Analysis of 2-NBDG uptake rate in cells treated with insulin (100 nM) or (α_2_M* 60 nM + insulin 100 nM) for 5 min at 37 °C. Simple linear regression analysis was performed by Infostat. * *p* < 0.05 indicates significant differences between insulin and α_2_M* + insulin.

**Figure 5 ijms-22-06915-f005:**
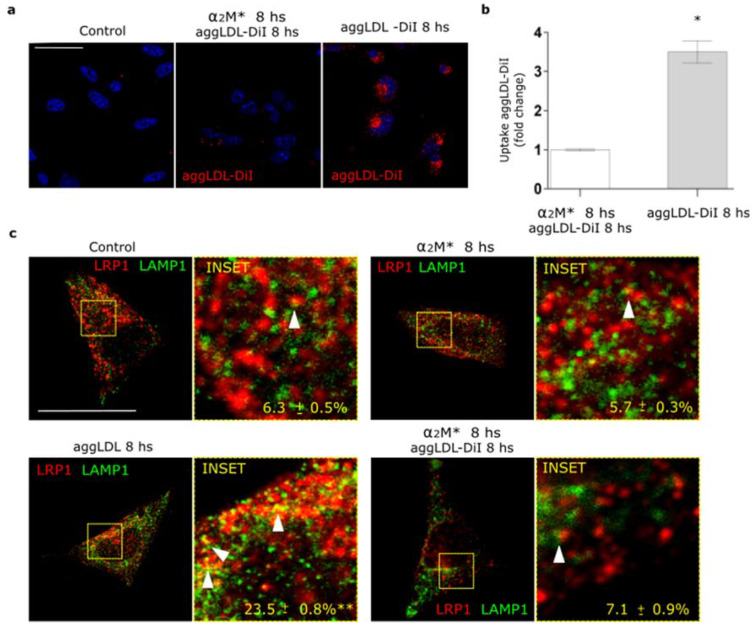
α_2_M* blocks the aggLDL intracellular accumulation and redistribution of LRP1 to degradation compartments in HL-1 cardiomyocytes. (**a**) Confocal microscopy in cells treated with α_2_M* (60 nM) or aggLDL-DiI (red) (100 µg/mL) or both ligands combined for 8 h at 37 °C. Hoechst was used to stain nuclei. Scale bar = 10 µm. (**b**) Quantitative analysis of aggLDL-DiI uptake as fluorescence intensity per cell area expressed as fold change with respect to the non-stimulated control. Values are expressed as mean ± SD. * *p* < 0.05 vs. indicated conditions. At least 20 cells per condition were analyzed (*n* = 20). (**c**) Confocal microscopy in cells treated with α_2_M* (60 nM) or aggLDL 100 µg/mL or both ligands combined for 8 h at 37 °C. Images show colocalization between LRP1 (red) and LAMP1 (green). INSET represents 4× magnification of the framed regions in dotted lines. White arrowheads indicate colocalization sectors. Scale bar = 10 µm. Quantitative analysis of colocalization between LRP1 and LAMP1 by Manders’ coefficients expressed as mean ± SD (%). * *p* < 0.05 vs. control conditions. At least 20 cells per condition were analyzed (*n* = 20).

**Figure 6 ijms-22-06915-f006:**
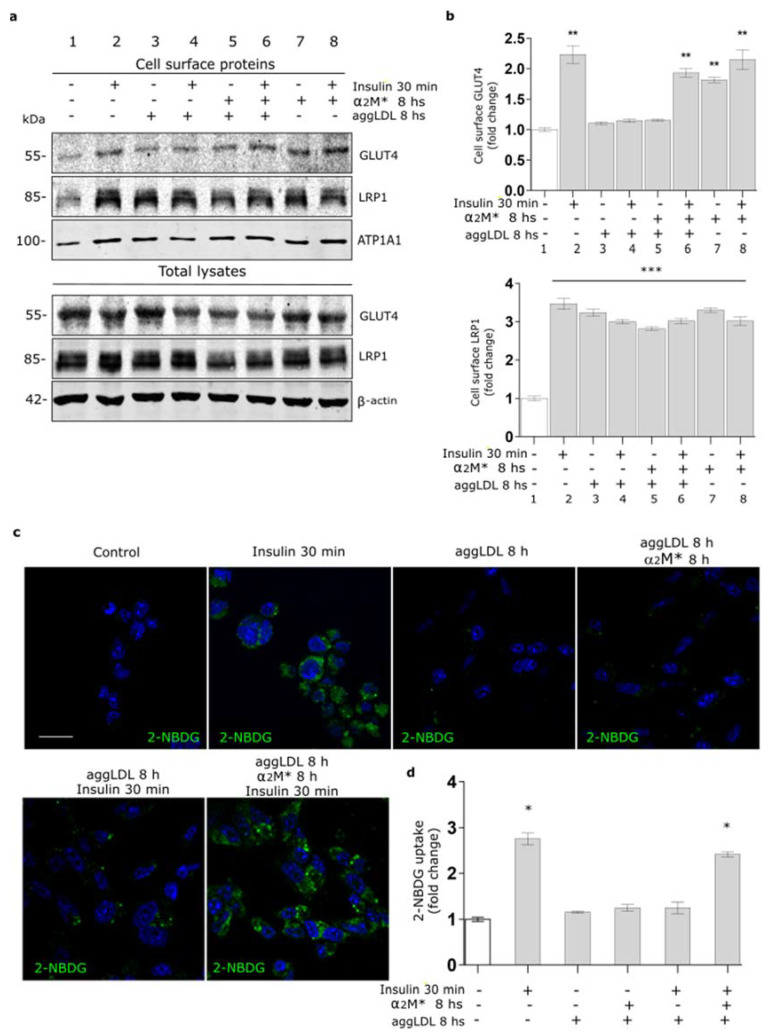
α_2_M* counteracts GLUT4 trafficking to PM and glucose uptake impaired by aggLDL in HL-1 cardiomyocytes. (**a**) Biotin-labeling protein assay to measure GLUT4 and LRP1 levels in the PM of cells stimulated with α_2_M* 60 nM, or aggLDL 100 µg/mL or both for 8 h, and then stimulated with insulin 100 nM for 30 min at 37 °C. Biotin-labeled proteins were isolated with streptavidin-conjugated beads and then analyzed by Western blot for GLUT4 and LRP1 expression. ATP1A1 and β-actin were used as protein loading controls. (**b**) Densitometric quantification of Western blot data for surface GLUT4 and LRP1 related to biotin-ATP1A1 expressed as fold change with respect to the non-stimulated control (white bar). Values are expressed as mean ± SEM. ** *p* < 0.01 and *** *p* < 0.001 vs. non-stimulated control. Three experiments were performed (*n* = 3). (**c**) Confocal microscopy of HL-1 cardiomyocytes treated with α_2_M* 60 nM, aggLDL 100 µg/mL or both for 8 h and then stimulated with insulin 100 nM together with 2-NBDG 80 µM (green) for 30 min at 37 °C. Hoechst was used to stain nuclei. Scale bar = 10 µm. (**d**) Quantitative analysis of 2-NBDG uptake as fluorescence intensity per cell area expressed as fold change with respect to the non-stimulated control. Values are expressed as mean ± SD. * *p* < 0.05 vs. non-stimulated control. At least 20 cells per condition were analyzed (*n* = 20).

**Figure 7 ijms-22-06915-f007:**
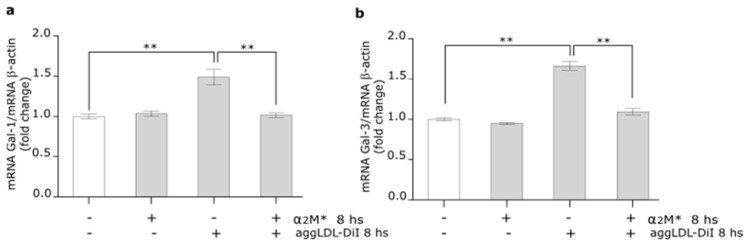
α_2_M* reduces Gal-1 and Gal-3 expression induced by aggLDL in HL-1 cardiomyocytes. (**a**,**b**) Quantitative RT-PCR assay to measure Gal-1 and Gal-3 mRNA expression of HL-1 cardiomyocytes treated with α_2_M* 60 nM, aggLDL 100 µg/mL or both ligands combined for 8 h at 37 °C. The bar graph shows the mean ± SEM of Gal-1 and Gal-3 mRNA levels relative to β-actin mRNA expressed as fold change respect to control. ** *p* < 0.01 vs. non-stimulated control (white bar).

**Figure 8 ijms-22-06915-f008:**
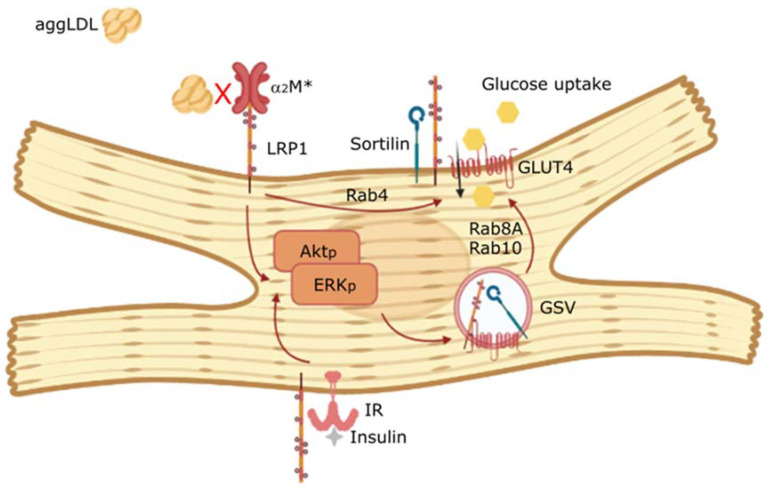
Schematic summary of the main findings. α_2_M* is an agonist ligand of LRP1 that improves the insulin response characterized by PI_3_K/Akt and MAPK/ERK signaling activation, GLUT4 translocation to PM by Rab4, Rab8A and Rab10 GTPases activation and glucose uptake induced by insulin in cardiomyocytes. Moreover, α_2_M* counteracts the antagonist effect of aggLDL on LRP1 in insulin response in lipid-loaded HL-1 cardiomyocytes.

## Data Availability

Not applicable.

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
