# Peer review of "Activated Alpha-2 Macroglobulin Improves Insulin Response via LRP1 in Lipid-Loaded HL-1 Cardiomyocytes"

_ijms, 2021, doi:10.3390/ijms22136915_

Round 1

Reviewer 1 Report

Dear Authors,

Your manuscript is well writen. I have question regarding your method, though. Did you perform replications for the Werstern blot analysis? How many replication? Thanks.

Best regards,

Author Response

Reviewer #1:

Your manuscript is well writen. I have question regarding your method, though. Did you perform replications for the Werstern blot analysis? How many replication? Thanks.

Response:

The Western blots were performed in three independent experiments, carrying out each condition individually in each of them (n = 3).

Reviewer 2 Report

Skrócenie bardzo obszernej literatury, odnoszÄ…cej siÄ™ gÅ‚ównie do pozycji z ostatnich 10 lat.

Author Response

Reviewer #2:

Skrócenie bardzo obszernej literatury, odnoszÄ…cej siÄ™ gÅ‚ównie do pozycji z ostatnich 10 lat. (Shortening of a very extensive literature, mainly relating to items from the last 10 years)

Response:

We appreciate the reviewer's comment but we consider that all the bibliography included is necessary for the adequate understanding and interpretation of this work, in addition, most of the references are from the last 10 years with the exception of specific publications related to specific methodologies or very well knew findings.

Reviewer 3 Report

The authors tested the effects of activated alpha-2 macroglobulin on aggLDL treated HL-1 cardiomyocytes.
They have previously showed that aggLDL exposed HL-1 cardiomyocytes promoted cholesteryl ester
accumulation and impaired their insulin response (doi:10.3390/cells9010182). Now, they showed that
activated alpha-2 macroglobulin counteracted these effects via the LRP1 and the signaling activation of PI3K and MAPK/ERK and the translocation of GLUT4 to the plasma membrane.

More comments:

The novelty and interest depends on the interest of the reader.
They 've studied retinal Muler glial cells (MIO-M1) along similar lines with the HL-1 cardiomyocytes but this cell line is a contracting cell-line of mouse atrial lineags. Therefore the new evidence
suggests changes that may occur in the heart. The authors could assesse HL-1 contractility however such an experiment was absent from their report in cells (10.3390/cells9010182).

An additional experiment would be to assess the contractility of HL-1 under aggLDL and activated alpha-2 macroglobulin conditions. 

Author Response

Reviewer #3:

The authors tested the effects of activated alpha-2 macroglobulin on aggLDL treated HL-1 cardiomyocytes. They have previously showed that aggLDL exposed HL-1 cardiomyocytes promoted cholesteryl ester accumulation and impaired their insulin response (doi:10.3390/cells9010182). Now, they showed that activated alpha-2 macroglobulin counteracted these effects via the LRP1 and the signaling activation of PI3K and MAPK/ERK and the translocation of GLUT4 to the plasma membrane.

More comments: The novelty and interest depends on the interest of the reader. They 've studied retinal Muler glial cells (MIO-M1) along similar lines with the HL-1 cardiomyocytes but this cell line is a contracting cell-line of mouse atrial lineags. Therefore the new evidence suggests changes that may occur in the heart. The authors could assesse HL-1 contractility however such an experiment was absent from their report in cells (10.3390/cells9010182). An additional experiment would be to assess the contractility of HL-1 under aggLDL and activated alpha-2 macroglobulin conditions.

Response:

We consider that this work is novel and provides interesting results regarding the role of α2M * not only to prevent the deleterious effect of the esterified cholesterol uptake in the myocardium, but also for the improvement in the insulin response without lipid overload, which it had not been proven until now. This could be beneficial not only in hypercholesterolemic conditions but also in patients with insulin resistance with a normal lipid profile. On the other hand, we agree with the reviewer that it would be interesting to evaluate cardiac contractility. Several recent reports have shown that cardiac steatosis is associated with lower heart contractibility, increased remodeling, and cardiac fibrosis. In this sense, we are currently conducting assays to evaluate functional aspects of HL-1 under treatment with aggLDL, insulin, and α2M *, which actually are including in the scope of a new study.